# FLAGIFYING THE DOWKER COMPLEX

## ABSTRACT

The Dowker complex $\mathrm{D}_R(X, Y)$ is a simplicial complex capturing the topological interplay between two finite sets $X$ and $Y$ under some relation $R \subseteq X \times Y$. While its definition is asymmetric, the famous Dowker duality states that $\mathrm{D}_R(X, Y)$ and $\mathrm{D}_R(Y, X)$ have homotopy equivalent geometric realizations. We introduce the Dowker-Rips complex $\mathrm{DR}_R(X, Y)$, defined as the flagification of the Dowker complex or, equivalently, as the maximal simplicial complex whose 1-skeleton coincides with that of $\mathrm{D}_R(X, Y)$. This is motivated by applications in topological data analysis, since as a flag complex, the Dowker-Rips complex is less expensive to compute than the Dowker complex. While the Dowker duality does not hold for Dowker-Rips complexes in general, we show that one still has that $\mathrm{H}_i(\mathrm{DR}_R(X, Y)) \cong \mathrm{H}_i(\mathrm{DR}_R(Y, X))$ for $i = 0, 1$. We further show that this weakened duality extends to the setting of persistent homology, and quantify the "failure" of the Dowker duality in homological dimensions higher than 1 by means of interleavings. This makes the Dowker-Rips complex a less expensive, approximate version of the Dowker complex that is usable in topological data analysis. Indeed, we provide a Python implementation of the Dowker-Rips complex and, as an application, we show that it can be used as a drop-in replacement for the Dowker complex in a tumor microenvironment classification pipeline. In that pipeline, using the Dowker-Rips complex leads to increase in speed while retaining classification performance.

## 1 INTRODUCTION

Topological data analysis (TDA) provides a framework for extracting qualitative geometric and topological features from complex data sets. Central to this approach is the construction of simplicial complexes that approximate the shape of an data set or, more generally, a metric space. A prominent example of such a complex is the Čech complex, where a finite set of points is declared to span a simplex precisely if the balls of some fixed radius $\varepsilon > 0$ around the points have non-empty intersection. While the Čech complex provably captures the topology of the union of all $\varepsilon$-balls, it is notoriously expensive to compute because triple and higher order intersections of balls must be checked (see, e.g., Ghrist (2014, Chapter 2.5) and Edelsbrunner & Harer (2010, Chapter III)). As a way around this, one often resorts to working with a simpler complex known as the Vietoris-Rips complex in practice. By definition, the Vietoris-Rips complex is obtained by flagifying of the Čech complex, that is, by adding all possible simplices whose edges are already present in the Čech complex. By construction, the Vietoris-Rips complex is thus entirely determined by its 1-skeleton, which coincides with that of the Čech complex. This makes the Vietoris-Rips complex less expensive to describe, compute and store. Indeed, several software packages for computing persistent homology like GUDHI (Maria, 2023) and ripser (Bauer, 2021) allow for a significant speed-up in computation time when working with flag complexes. Moreover, even though the Vietoris-Rips complex does not enjoy the same theoretical guarantees regarding the capturing of the topology of the underlying data set, it is guaranteed to be "topologically close" to the Čech complex in the sense that the two complexes are interleaved. Finally, there do exist conditions under which such guarantees for the Vietoris-Rips complex do exist (Chambers et al., 2010; Attali et al., 2013).

While both the Čech and Vietoris-Rips complexes are used to analyze a single data set, one might be interested in analyzing the topology of a data set relative to another one living in the same space (or,

equivalently, the topology of a subset of a data set relative to its complement). One tool for doing so is the Dowker complex, which was introduced by Dowker in 1952 Dowker (1952).

**Definition 1.1.** *Let $X, Y$ be two finite sets and let $R \subseteq X \times Y$ be a non-empty relation. The* Dowker *complex on $X$ relative to $Y$ is the simplicial complex $\mathrm{D}_R(X, Y)$ defined by the rule that a finite subset $\sigma \subseteq X$ belongs to $\mathrm{D}_R(X, Y)$ iff there exists $y \in Y$ such that $(x, y) \in R$ for all $x \in \sigma$.*

If $X$ and $Y$ in Definition 1.1 are subsets of a metric space $(Z, d)$, one may define a relation $R_\varepsilon \subseteq X \times Y$ by declaring $(x, y) \in R_\varepsilon$ iff $d(x, y) \leq \varepsilon$ for $\varepsilon \geq 0$. In this setting, the Dowker complex may be regarded as a variant of the Čech complex where one does not simply require the intersection of $\varepsilon$-balls around elements of $X$ to be non-empty, but indeed to contain an element of $Y$.

A particularly nice feature of the Dowker complex is given by the *Dowker duality*, proven by Dowker in the original paper introducing Dowker complexes (Dowker, 1952). It states that the two complexes $\mathrm{D}_R(X, Y)$ and $\mathrm{D}_R(Y, X)$ are homotopy equivalent and, as a consequence, have isomorphic homology groups. This result has been extended to filtrations of Dowker complexes by Chowdhury and Mémoli, who have shown that these homotopy equivalences commute with the inclusions of the filtrations, thus extending Dowker duality to the setting of persistent homology (Chowdhury & Mémoli, 2018). In other words, this more general form of Dowker duality allows one to compute persistent homology for an entire filtration of Dowker complexes $\{\mathrm{D}_R(X, Y)\}_{R \in \mathcal{R}}$ for some set $\mathcal{R}$ of nested relations, and this persistent homology is guaranteed to be isomorphic to that of the corresponding filtration $\{\mathrm{D}_R(Y, X)\}_{R \in \mathcal{R}}$. In particular, this may be applied to the relations $R_\varepsilon$ in the setting of metric spaces. From a practical perspective, this duality allows one to compute the smaller of the two complexes at each step (which amounts to potentially swapping the roles of $X$ and $Y$). This can be crucial for computation time and memory consumption, in particular if one of $X$ and $Y$ is significantly smaller than the other. In the context of metric spaces, the persistence diagrams resulting from filtrations of Dowker complexes provide a way of analyzing whether and how the classes $X$ and $Y$ are colocalized in the ambient metric space $Z$ (see, e.g., Stolz et al. (2024, Section 5.1.2) for details). Dowker complexes have seen applications inside math as well as outside of math, in domains as diverse as computational biology, data science, machine learning and neuroscience (Stolz et al., 2024; Choi et al., 2024; Brun & Blaser, 2019; Zemene & Pelillo, 2015; Liu et al., 2022; Moshkov et al., 2022; Vaupel et al., 2023; Freund et al., 2015; Garland et al., 2016). For more details on Dowker complexes, see, e.g., Chazal et al. (2014); Ghrist (2014); Chowdhury & Mémoli (2018).

In this work, we introduce and examine a flagified version of the Dowker complex, which we call the *Dowker-Rips complex*. Just like the Vietoris-Rips complex may be defined as a flagified version of the Čech complex and can thus be regarded as a less expensive and approximate variant thereof, the Dowker-Rips complex can be regarded as such a variant of the Dowker complex. To define the Dowker-Rips complex, we first state a precise definition of flagifications.

**Definition 1.2.** *Given a simplicial complex $X$, the* flagification *of $X$, denoted by $\mathcal{F}(X)$, is defined as the simplicial complex that is obtained from $X$ by including a simplex $\sigma \subseteq X$ whenever all edges of $\sigma$ already belong to $X$ and $\dim(\sigma) \geq 2$. More generally, for an integer $k \geq 2$, the $k$-flagification of $X$, denoted by $\mathcal{F}^{\geq k}(X)$, is defined as the complex that is obtained from $X$ by including a simplex $\sigma \subseteq X$ whenever all $(k-1)$-dimensional faces of $\sigma$ already belong to $X$ and $\dim(\sigma) \geq k$.*

**Remark 1.3.** *Note that $X \subseteq \mathcal{F}^{\geq k}(X) \subseteq \mathcal{F}(X)$ for any simplicial complex $X$ and $k \geq 2$. Moreover, we have that $X = \mathcal{F}^{\geq k}(X)$ if $k > \dim(X) + 1$, and $\mathcal{F}^{\geq 2}(X) = \mathcal{F}(X)$ for any simplicial complex $X$. Finally, note that $\mathcal{F}^{\geq k}(X)$ is determined entirely by the $(k-1)$-skeleton of $X$, $k \geq 2$.*

**Example 1.4.** *Let $X \subseteq \mathbb{R}^n$, and denote by $\check{\mathrm{C}}_\varepsilon(X)$ and $\mathrm{VR}_\varepsilon(X)$ its Čech and Vietoris-Rips complexes at some scale $\varepsilon \geq 0$, respectively. Then we have that $\mathcal{F}(\check{\mathrm{C}}_\varepsilon(X)) = \mathrm{VR}_\varepsilon(X)$.*

With the definition of flagification at hand, we are now ready to define the Dowker-Rips complex.

**Definition 1.5.** *Let $X, Y$ be two finite sets and let $R \subseteq X \times Y$ be a non-empty relation. The* Dowker-Rips *complex on $X$ relative to $Y$ is defined as*

$$\mathrm{DR}_R(X, Y) \coloneqq \mathcal{F}(\mathrm{D}_R(X, Y)).$$

The motivation behind defining the Dowker-Rips complex is twofold. First, the Dowker complex is a Čech-like complex in the sense that its construction relies on the pairwise and higher order intersections

of metric balls around its elements containing a certain element. From a theoretical perspective, it thus seems natural to define a complex that relates to the Dowker complex in the same way as the Vietoris-Rips complex relates to the Čech complex, namely through flagification. Second, from a practical perspective, the Dowker complex (and its persistent homology) is prohibitively expensive to compute for large or high-dimensional data sets. The Dowker-Rips complex provides an alternative to the Dowker complex that is applicable in practice, while at the same time retaining the usefulness of the latter. In this work, we provide a theoretical analysis of the differences between the Dowker and the Dowker-Rips complexes, and we illustrate the usefulness of the latter by showing that simply replacing the Dowker complex with the Dowker-Rips complex in an existing tumor microenvironment classification pipeline leads to increase in speed while retaining classification performance.

Given the definition of the Dowker-Rips complex, there are two natural questions that arise:

(1) How much can the Dowker-Rips complex differ from the Dowker complex?

(2) Does some version of the Dowker duality still hold for Dowker-Rips complexes?

For filtrations of simplicial complexes, questions such as Question (1) are usually answered by showing that the two filtrations are *multiplicatively c-interleaved* for some $c \geq 1$.[1] Informally speaking, the smaller the value of $c \geq 1$, the closer the two filtrations are. A prominent example of this is the chain of inclusions

$$\check{C}_\varepsilon(X) \subseteq \mathrm{VR}_\varepsilon(X) \subseteq \check{C}_{2\varepsilon}(X) \tag{1}$$

for $\varepsilon \geq 0$, which translates into the fact that the Vietoris-Rips complex and the Čech complex are multiplicatively 2-interleaved. We show that a similar argument also works for Dowker-Rips and Dowker complexes in the case where $X$ and $Y$ are subsets of some metric space $(Z, d)$ with the relation $R_\varepsilon \subseteq X \times Y$ defined by declaring $(x, y) \in R$ iff $d(x, y) \leq \varepsilon$ for $\varepsilon \geq 0$.

**Theorem 1.6.** *Let $X, Y \subseteq Z$ where $(Z, d)$ is some metric space, and define the relations $R_\varepsilon \subseteq X \times Y$ by declaring $(x, y) \in R$ iff $d(x, y) \leq \varepsilon$ for $\varepsilon \geq 0$. Denote by $\mathrm{D}_\bullet(X, Y)$ the filtration given by $\{\mathrm{D}_{R_\varepsilon}(X, Y)\}_{\varepsilon \in \mathbb{R}^+}$, and similarly for $\mathrm{DR}_\bullet(X, Y)$. Then have that*

$$\mathrm{D}_\varepsilon(X, Y) \subseteq \mathrm{DR}_\varepsilon(X, Y) \subseteq \mathrm{D}_{3\varepsilon}(X, Y) \tag{2}$$

*for all $\varepsilon \geq 0$, and, in particular, that $\mathrm{D}_\bullet(X, Y)$ and $\mathrm{DR}_\bullet(X, Y)$ are multiplicatively 3-interleaved.*

The above result is sharp in the sense that the inclusion $\mathrm{DR}_\varepsilon(X, Y) \subseteq \mathrm{D}_{3\varepsilon}(X, Y)$ does not hold when 3 is replaced by some value $c < 3$ (see Proposition 3.1 for such an example).

We use a similar argument to give a partial answer to Question (2). We point out that the multiplicative interleaving claimed in the following does not stem from a chain of inclusions such as in Equations (1) and (2), but rather from the more general notion of a multiplicative interleaving defined in Section 3.

**Theorem 1.7.** *Let $X, Y \subseteq Z$ where $(Z, d)$ is some metric space, and define the relations $R_\varepsilon \subseteq X \times Y$ as in Theorem 1.6, $\varepsilon \geq 0$. Denote by $\mathrm{DR}_\bullet(X, Y)$ the filtration given by $\{\mathrm{DR}_{R_\varepsilon}(X, Y)\}_{\varepsilon \in \mathbb{R}^+}$, and similarly for $\mathrm{DR}_\bullet(Y, X)$. Then $\mathrm{DR}_\bullet(X, Y)$ and $\mathrm{DR}_\bullet(Y, X)$ are multiplicatively 3-interleaved.*

While this already establishes that $\mathrm{DR}_\bullet(X, Y)$ and $\mathrm{DR}_\bullet(Y, X)$ cannot be "too different", it is still a significantly weaker guarantee than the one we have for Dowker complexes, where we have a homotopy equivalence and thus an isomorphism at the level of persistent homology. Indeed, as we will see in Section 4, an isomorphism at the level of persistent homologies of $\mathrm{DR}_\bullet(X, Y)$ and $\mathrm{DR}_\bullet(Y, X)$ does not exist in general. Nevertheless, we still obtain an isomorphism at the level of persistent homology when restricted to homological dimensions 0 and 1. This follows from a slightly more general result on $k$-flagifications of Dowker complexes.

**Theorem 1.8.** *Let $X$ and $Y$ be two finite sets and let $\{R_j\}_{j \in J}$ be a sequence of relations such that $R_j \subseteq X \times Y$ for all $j \in J$, and $R_j \subseteq R_{j'}$ whenever $j \leq j'$, where $J$ is some totally ordered index set. Given an*

---

[1]For the definition of a multiplicative interleaving, see Definition 2.2.

*integer $k \geq 2$, denote by $\mathcal{F}^{\geq k}(\mathrm{D}_\bullet(X,Y))$ the filtration given by $\left\{\mathcal{F}^{\geq k}(\mathrm{D}_{R_j}(X,Y))\right\}_{j \in J}$, and similarly for $\mathcal{F}^{\geq k}(\mathrm{D}_\bullet(Y,X))$. Then we have that*

$$\mathrm{PH}_i(\mathcal{F}^{\geq k}(\mathrm{D}_\bullet(X,Y))) \cong \mathrm{PH}_i(\mathcal{F}^{\geq k}(\mathrm{D}_\bullet(Y,X)))$$

*for $i = 0, \ldots, k-1$.*

**Remark 1.9.** *Recall that for large enough $k \geq 1$, we have that $\mathcal{F}^{\geq k}(\mathrm{D}_R(X,Y)) = \mathrm{D}_R(X,Y)$ and $\mathcal{F}^{\geq k}(\mathrm{D}_R(Y,X)) = \mathrm{D}_R(Y,X)$. For such choices of $k$, Theorem 1.8 is essentially a homological (and hence weaker) restatement of Chowdhury & Mémoli (2018, Theorem 3). Indeed, Theorem 1.8 may be read as saying that there exists a decreasing sequence of filtrations*

$$\mathrm{DR}_\bullet(X,Y) = \mathcal{F}^{\geq 2}(\mathrm{D}_\bullet(X,Y)) \supseteq \cdots \supseteq \mathcal{F}^{\geq k}(\mathrm{D}_\bullet(X,Y)) \supseteq \mathcal{F}^{\geq k'}(\mathrm{D}_\bullet(X,Y)) \supseteq \cdots \supseteq \mathrm{D}_\bullet(X,Y)$$

*for $k < k'$, in which the number of dimensions for which Dowker duality holds increases by $1$ at each step.*

Using the fact that the Dowker-Rips complex is the 2-flagification of the Dowker complex, we get the following *Dowker-Rips duality*.

**Theorem 1.10.** *Let $(Z,d)$ be a metric space and let $X, Y \subseteq Z$ be non-empty and finite disjoint subsets. For $\varepsilon \geq 0$, define the relation $R_\varepsilon \subseteq X \times Y$ by*

$$(x,y) \in R_\varepsilon \quad \textit{iff} \quad d(x,y) \leq \varepsilon.$$

*Denote by $\mathrm{DR}_\bullet(X,Y)$ the filtration given by $\{\mathrm{DR}_{R_\varepsilon}(X,Y)\}_{\varepsilon \in \mathbb{R}^+}$, and similarly for $\mathrm{DR}_\bullet(Y,X)$. Then we have that*

$$\mathrm{PH}_i(\mathrm{DR}_\bullet(X,Y)) \cong \mathrm{PH}_i(\mathrm{DR}_\bullet(Y,X))$$

*for $i = 0, 1$.*

The above result is sharp in the sense that its conclusion does not hold for homological dimensions higher than 1 (see Proposition 4.4 for such an example). Nevertheless, the Dowker-Rips duality is a desirable property of the Dowker-Rips complex, since, in practice, persistent homology is often computed only up to homological dimension 1 for reasons of computational complexity. In these homological dimensions, the Dowker-Rips duality may thus be used to accelerate the computation of the persistent homology of the Dowker-Rips complex: like in the case of the Dowker complex, this duality allows one to potentially swap the roles of $X$ and $Y$ in order to compute the less expensive variant of the two Dowker-Rips complexes.

This paper is organized as follows. In Section 2, we briefly review the necessary mathematical background. In Section 3, we construct the multiplicative interleavings, proving Theorems 1.6 and 1.7. In Section 4, which is the main technical section, is devoted to deducing the Dowker-Rips duality (Theorem 1.10). Finally, in Section 5, we present the application that justifies using the Dowker-Rips complex instead of the Dowker complex in practice.

## 2 PRELIMINARIES

In this section, we briefly review the necessary background on the concepts and tools stemming from topological data analysis (TDA) used in this paper. We refer the reader to Schnider et al. (2025); Edelsbrunner & Harer (2010); Ghrist (2014) for details on the following.

### 2.1 SIMPLICIAL COMPLEXES AND FILTRATIONS

A *simplicial complex* is a combinatorial structure that can be seen as a higher-dimensional generalization of a graph. Formally, it is a collection $K$ of finite subsets of some vertex set $X$ such that if $\sigma \in K$ and $\tau \subseteq \sigma$, then $\tau \in K$. Each subset $\sigma \subseteq X$ belonging to $K$ is called a *simplex*, and usually denoted by $\sigma = [x_0, \ldots, x_n]$, where $x_1, \ldots, x_n \in X$. The *dimension* of a simplex $\sigma$ is defined as $\dim(\sigma) := |\sigma| - 1$. Simplices of dimension 0 and 1 are also referred to as *vertices* and *edges*, respectively.

A *filtration* of a topological space $X$ is a nested sequence of subspaces

$$X_{i_0} \subseteq X_{i_1} \subseteq \cdots \subseteq X_{i_n} = X,$$

for some $i_0 \leq i_1 \leq \cdots \leq i_n \in I$, where $I$ is some totally ordered index set. Such a filtration may be succinctly written as $X_\bullet = \{X_{i_k}\}_{k \geq 0}$. In TDA, we typically have that $I = \mathbb{R}$, and that the filtration indices represent some scale parameter, as is the case in the following example.

**Example 2.1.** *Given a metric space $(Z, d)$ and a subset $X \subseteq Z$, the* Čech complex of $X$ at scale $\varepsilon \geq 0$, *denoted by $\check{C}_\varepsilon(X, Z)$, is the simplicial complex defined as containing a simplex $[x_0, \ldots, x_k] \subseteq X$ if the closed $\varepsilon$-balls centered at $x_0, \ldots, x_k$ have a non-empty common intersection in $Z$. If $Z = \mathbb{R}^n$, one usually writes $\check{C}_\varepsilon(X)$ instead of $\check{C}_\varepsilon(X, \mathbb{R}^n)$. In contrast, the* Vietoris-Rips complex of $X$ at scale $\varepsilon \geq 0$, *denoted by $\mathrm{VR}_\varepsilon(X)$, is defined as the simplicial complex containing a simplex $[x_0, \ldots, x_k] \subseteq X$ if $d(x_i, x_j) \leq 2\varepsilon$ for all $0 \leq i \leq j \leq k$. Both complexes induce filtrations $\check{C}_\bullet(X, Z) := \left\{ \check{C}_\varepsilon(X, Z) \right\}_{\varepsilon \in \mathbb{R}^+}$ and $\mathrm{VR}_\bullet(X) := \left\{ \mathrm{VR}_\varepsilon(X) \right\}_{\varepsilon \in \mathbb{R}^+}$, obtained by gradually increasing the value of the scale parameter $\varepsilon$.*

## 2.2 Persistent homology and persistence modules

*Persistent homology* (PH) formalizes the study of topological features across a filtration. For each $k \geq 0$, PH keeps track of the $k$-th homology group across the evolution of a filtration. More formally, given a filtration $X_\bullet$, this is achieved by applying the $k$-dimensional homology functor to the sequence of inclusion maps

$$X_{i_0} \hookrightarrow X_{i_1} \hookrightarrow \cdots \hookrightarrow X_{i_n} = X.$$

This yields a collection of vector spaces

$$\mathrm{H}_k(X_{i_0}) \to \mathrm{H}_k(X_{i_1}) \to \cdots \to \mathrm{H}_k(X_{i_n})$$

with induced maps between them. This data is denoted by $\mathrm{PH}_k(X_\bullet)$ and an example of a *persistence module*. In general, the latter is defined as any indexed collection of vector spaces $\mathbb{V} = \{V_i\}_{i \in I}$ (for some totally ordered set $I$) with linear maps $f_{i,j} \colon V_i \to V_j$, $i \leq j$, such that $f_{i,i} = \mathrm{id}_{V_i}$ and $f_{i,k} = f_{j,k} \circ f_{i,j}$ for any $i \leq j \leq k \in I$. Two persistence modules $\{V_i\}_{i \in I}$ and $\{W_i\}_{i \in I}$ are said to be isomorphic *isomorphic* if there exists a collection of isomorphisms $\varphi_i \colon V_i \to W_i$, $i \in I$, such that the diagrams

$$
\begin{array}{ccc}
V_i & \longrightarrow & V_j \\
\varphi_i \downarrow & & \downarrow \varphi_j \\
W_i & \longrightarrow & W_j
\end{array}
\quad \text{and} \quad
\begin{array}{ccc}
V_i & \longrightarrow & V_j \\
\varphi_i^{-1} \uparrow & & \uparrow \varphi_j^{-1} \\
W_i & \longrightarrow & W_j
\end{array}
$$

commute.

## 2.3 Multiplicative interleavings

Interleavings are a way to capture similarities of filtrations. While in many cases additive interleavings are desirable, in some cases *multiplicative interleavings* are the best that can be done. Following we recall the definition of a multiplicative interleaving (see, e.g., Dey & Wang (2022); Oudot (2015)).

**Definition 2.2.** *Let $\mathcal{F} = \{F_a\}_{a \in \mathbb{R}}$ and $\mathcal{G} = \{G_a\}_{a \in \mathbb{R}}$ be filtrations. We say that $\mathcal{F}$ and $\mathcal{G}$ are* multiplicatively $c$-interleaved *if there are maps $\varphi_a \colon F_a \to G_{ca}$ and $\psi_a \colon G_a \to F_{ca}$ such that the following diagrams commute for every $a \in \mathbb{R}$ and $\varepsilon > 0$:*

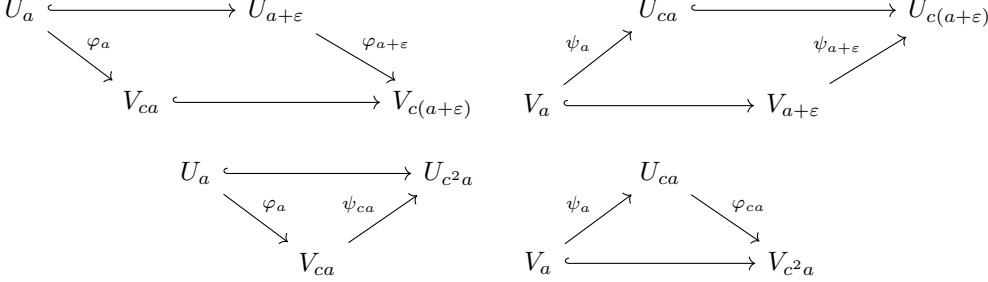

Note that the smaller the value of $c \geq 1$, the "closer" the two filtrations are to each other. As with additive interleavings, a multiplicative interleaving of two filtrations implies that the respective persistent homologies are "close" in a suitable sense. Multiplicative interleavings thus provide a rigorous way of quantifying how different two filtrations are.

One prominent example of a multiplicative interleaving stems from the chain of inclusions

$$\check{C}_\varepsilon(X) \subseteq \mathrm{VR}_\varepsilon(X) \subseteq \check{C}_{2\varepsilon}(X)$$

for $\varepsilon \geq 0$, which establishes a multiplicative 2-interleaving of the Čech filtration and the Vietoris-Rips filtration.

## 3   MULTIPLICATIVE INTERLEAVINGS OF THE DOWKER-RIPS COMPLEX

This section pertains to the two multiplicative interleavings whose existence was claimed in Section 1. For convenience, we restate the relevant theorems, and we refer the reader to Appendix A.1.1 for the proofs of the technical results of this section.

**Theorem 1.6.** *Let $X, Y \subseteq Z$ where $(Z, d)$ is some metric space, and define the relations $R_\varepsilon \subseteq X \times Y$ by declaring $(x, y) \in R$ iff $d(x, y) \leq \varepsilon$ for $\varepsilon \geq 0$. Denote by $\mathrm{D}_\bullet(X, Y)$ the filtration given by $\{\mathrm{D}_{R_\varepsilon}(X, Y)\}_{\varepsilon \in \mathbb{R}^+}$, and similarly for $\mathrm{DR}_\bullet(X, Y)$. Then have that*

$$\mathrm{D}_\varepsilon(X, Y) \subseteq \mathrm{DR}_\varepsilon(X, Y) \subseteq \mathrm{D}_{3\varepsilon}(X, Y) \tag{2}$$

*for all $\varepsilon \geq 0$, and, in particular, that $\mathrm{D}_\bullet(X, Y)$ and $\mathrm{DR}_\bullet(X, Y)$ are multiplicatively 3-interleaved.*

**Theorem 1.7.** *Let $X, Y \subseteq Z$ where $(Z, d)$ is some metric space, and define the relations $R_\varepsilon \subseteq X \times Y$ as in Theorem 1.6, $\varepsilon \geq 0$. Denote by $\mathrm{DR}_\bullet(X, Y)$ the filtration given by $\{\mathrm{DR}_{R_\varepsilon}(X, Y)\}_{\varepsilon \in \mathbb{R}^+}$, and similarly for $\mathrm{DR}_\bullet(Y, X)$. Then $\mathrm{DR}_\bullet(X, Y)$ and $\mathrm{DR}_\bullet(Y, X)$ are multiplicatively 3-interleaved.*

We conclude this section by providing an example illustrating that the interleaving from Theorem 1.6 is sharp in the sense that the inclusion $\mathrm{DR}_\varepsilon(X, Y) \subseteq \mathrm{D}_{3\varepsilon}(X, Y)$ does not hold when 3 is replaced by some value $c < 3$.

**Proposition 3.1.** *There exists a setting for Theorem 1.6 such that*

$$\mathrm{DR}_\varepsilon(X, Y) \not\subseteq \mathrm{D}_{c\varepsilon}(X, Y)$$

*for any $c < 3$.*

*Proof.* Define $(Z, d)$ as the graph pictured in Figure 1 equipped with the shortest-path metric, and let $X = \{x_0, x_1, x_2\} \subseteq Z$ and $Y = \{y_0, y_1, y_2\} \subseteq Z$ be the set of the crossed and hollow circles, respectively. It is easy to see that $[x_i, x_j] \in \mathrm{D}_1(X, Y)$ for all $0 \leq i < j \leq 2$, and hence that $[x_0, x_1, x_2] \in \mathrm{DR}_1(X, Y)$. In contrast, for $\mathrm{D}_c(X, Y)$, $c \geq 1$, to contain $[x_0, x_1, x_2]$, $c$ must be large enough to guarantee the existence of an element $y \in Y$ such that $d(y, x_i) \leq c$ for all $0 \leq i \leq 2$. Since $d(y_i, x_i) = 3$ for all $0 \leq i \leq 2$, this is the case only if $c \geq 3$. $\square$

## 4   DOWKER-RIPS DUALITY

In this section, we derive the strengthenings of the interleaving results from Section 3 and, in particular, the Dowker-Rips duality. We refer the reader to Appendix A.1.2 for the proofs of the technical results of this section. To begin, we restate and extend the definition of $k$-flagification to include a notion of partial flagification that is needed in the proofs.

**Definition 4.1.** *Given a simplicial complex $X$, the* flagification *of $X$, denoted by $\mathcal{F}(X)$, is defined as the simplicial complex that is obtained from $X$ by including a simplex $\sigma \subseteq X$ whenever all edges of $\sigma$ already belong to $X$ and $\dim(\sigma) \geq 2$. More generally, for an integer $k \geq 2$, the $k$-flagification of $X$, denoted by $\mathcal{F}^{\geq k}(X)$, is defined as the complex that is obtained from $X$ by including a simplex $\sigma \subseteq X$ whenever all $(k-1)$-dimensional faces of $\sigma$ already belong to $X$ and $\dim(\sigma) \geq k$. Finally, the* partial $k$-flagification *of $X$, denoted by $\mathcal{F}^k(X)$, is defined as the complex that is obtained from $X$ by including a simplex $\sigma \subseteq X$ whenever all $(k-1)$-dimensional faces of $\sigma$ already belong to $X$ and $\dim(\sigma) = k$.*

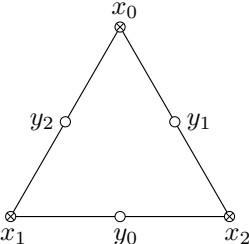

Figure 1: The metric space $(Z, d)$ from the proof of
Proposition 3.1, with subsets $X$ and $Y$ consisting of
the crossed and hollow circles, respectively.

Recall from Chowdhury & Mémoli (2018, Section 5.1) that there exists a simplicial map $\Gamma\colon \mathrm{D}_R^{(1)}(X, Y) \to \mathrm{D}_R(Y, X)$ that induces a homotopy equivalence $\psi\colon |\mathrm{D}_R^{(1)}(X, Y)| \to |\mathrm{D}_R(Y, X)|$ on the level of geometric realizations. Here and in what follows, $X^{(1)}$ denotes the first barycentric subdivision of a simplicial complex $X$. The map $\Gamma$ is defined by mapping any vertex $\sigma = [x_0, \ldots, x_n] \in \mathrm{D}_R^{(1)}(X, Y)$, $x_0, \ldots, x_n \in X$, to an element $y_\sigma \in Y$ such that $(x_k, y_\sigma) \in R$ for all $k = 0, \ldots, n$. It is shown in Chowdhury & Mémoli (2018) that the map $\Gamma$ thus defined is simplicial and, moreover, that different choices of $y_\sigma$ in its definition result in maps that are contiguous to one another (and hence induce homotopic maps on the level of geometric realizations). At a high level, we prove Theorem 1.10 by first showing in Lemma 4.2 that the map $\psi$ can be extended to a map between the partial $k$-flagifications. From this we deduce Proposition 4.3, the main technical result that establishes properties of the extensions of $\psi$ pertaining to homology and commutativity. Finally, Theorems 1.8 and 1.10 will be relatively straight forward consequences of that proposition.

To make sense of the setup of Lemma 4.2, observe that $\mathrm{D}_R(X, Y)$ is a subcomplex of $\mathcal{F}^k(\mathrm{D}_R(X, Y))$, which implies that $\mathrm{D}_R^{(1)}(X, Y)$ is a subcomplex of $\mathcal{F}^k(\mathrm{D}_R(X, Y))^{(1)}$ for $k \geq 2$.

**Lemma 4.2.** *The homotopy equivalence $\psi\colon |\mathrm{D}_R^{(1)}(X, Y)| \to |\mathrm{D}_R(Y, X)|$ extends to a continuous map*

$$\varphi\colon |\mathcal{F}^k(\mathrm{D}_R(X, Y))^{(1)}| \to |\mathcal{F}^k(\mathrm{D}_R(Y, X))|$$

*for any $k \geq 2$.*

With the previous lemma at hand, we can now deduce the required properties of the extensions of the map $\psi$.

**Proposition 4.3.** *Let $X$ and $Y$ be two finite sets, let $R \subseteq R' \subseteq X \times Y$ be two non-empty relations, and let $k \geq 2$ an integer. Then there exist continuous maps $\varphi\colon |\mathcal{F}^k(\mathrm{D}_R(X, Y))| \to |\mathcal{F}^k(\mathrm{D}_R(Y, X))|$ and $\varphi'\colon |\mathcal{F}^k(\mathrm{D}_{R'}(X, Y))| \to |\mathcal{F}^k(\mathrm{D}_{R'}(Y, X))|$ that induce isomorphisms on the level of $i$-dimensional homology for $i = 0, \ldots, k - 1$, and, moreover, such that the diagram*

$$
\begin{array}{ccc}
|\mathcal{F}^k(\mathrm{D}_R(X, Y))| & \longhookrightarrow & |\mathcal{F}^k(\mathrm{D}_{R'}(X, Y))| \\
\varphi \downarrow & & \downarrow \varphi' \\
|\mathcal{F}^k(\mathrm{D}_R(Y, X))| & \longhookrightarrow & |\mathcal{F}^k(\mathrm{D}_{R'}(Y, X))|
\end{array}
\tag{3}
$$

*commutes up to homotopy. Here, the horizontal maps are given by inclusion.*

The proposition above allow us to prove the main theorems, which we restate for convenience.

**Theorem 1.8.** *Let $X$ and $Y$ be two finite sets and let $\{R_j\}_{j \in J}$ be a sequence of relations such that $R_j \subseteq X \times Y$ for all $j \in J$, and $R_j \subseteq R_{j'}$ whenever $j \leq j'$, where $J$ is some totally ordered index set. Given an integer $k \geq 2$, denote by $\mathcal{F}^{\geq k}(\mathrm{D}_\bullet(X, Y))$ the filtration given by $\{\mathcal{F}^{\geq k}(\mathrm{D}_{R_j}(X, Y))\}_{j \in J}$, and similarly for $\mathcal{F}^{\geq k}(\mathrm{D}_\bullet(Y, X))$. Then we have that*

$$\mathrm{PH}_i(\mathcal{F}^{\geq k}(\mathrm{D}_\bullet(X, Y))) \cong \mathrm{PH}_i(\mathcal{F}^{\geq k}(\mathrm{D}_\bullet(Y, X)))$$

*for $i = 0, \ldots, k - 1$.*

**Theorem 1.10.** *Let $(Z, d)$ be a metric space and let $X, Y \subseteq Z$ be non-empty and finite disjoint subsets. For $\varepsilon \geq 0$, define the relation $R_\varepsilon \subseteq X \times Y$ by*

$$(x, y) \in R_\varepsilon \quad \textit{iff} \quad d(x, y) \leq \varepsilon.$$

*Denote by $\mathrm{DR}_\bullet(X, Y)$ the filtration given by $\{\mathrm{DR}_{R_\varepsilon}(X, Y)\}_{\varepsilon \in \mathbb{R}^+}$, and similarly for $\mathrm{DR}_\bullet(Y, X)$. Then we have that*

$$\mathrm{PH}_i(\mathrm{DR}_\bullet(X, Y)) \cong \mathrm{PH}_i(\mathrm{DR}_\bullet(Y, X))$$

*for $i = 0, 1$.*

We conclude this section by providing an example illustrating that the Dowker-Rips duality is sharp in the sense that its conclusion does not hold for homological dimensions higher than $1$.

**Proposition 4.4.** *There exists a setting for Theorem 1.10 in which the conclusion fails for $i = 2$.*

*Proof.* Let $X = \{x_0, \ldots, x_3\} \subseteq \mathbb{R}^3$ denote the set of vertices of a regular tetrahedron with edge length $1$ embedded in $\mathbb{R}^3$, and let $Y = \{y_{ij} \mid 0 \leq i < j \leq 3\}$, where $y_{ij}$ is defined to be the midpoint of $x_i$ and $x_j$, $0 \leq i < j \leq 3$. Denote by $\mathrm{D}_\bullet(X, Y)$ the filtration given by $\{\mathrm{D}_{R_\varepsilon}(X, Y)\}_{\varepsilon \in \mathbb{R}^+}$, and similarly for $\mathrm{D}_\bullet(Y, X)$. Then we have that $\mathrm{D}_{1/2}(X, Y)$ is homeomorphic to the geometric realization of $K_4$, the complete graph on four vertices. In contrast, the complex $\mathrm{D}_{1/2}(Y, X)$ has vertex set $Y$, and a set of vertices spans a simplex precisely when their subscripts share a common element. See Figure 2 for an illustration of the complexes $\mathrm{D}_{1/2}(X, Y)$ and $\mathrm{D}_{1/2}(Y, X)$.

It follows that the flagifications of $\mathrm{D}_{1/2}(X, Y)$ and $\mathrm{D}_{1/2}(Y, X)$ equal a 3-simplex and an octahedron, respectively. Hence $\mathrm{DR}_{1/2}(X, Y)$ and $\mathrm{DR}_{1/2}(Y, X)$ are homotopy equivalent to a point and a 2-sphere, respectively. This implies that

$$\mathrm{H}_2(\mathrm{DR}_{1/2}(X, Y)) \cong \{0\} \quad \text{and} \quad \mathrm{H}_2(\mathrm{DR}_{1/2}(Y, X)) \cong \mathbb{Z},$$

and, in particular, that

$$\mathrm{PH}_2(\mathrm{DR}_{1/2}(X, Y)) \ncong \mathrm{PH}_2(\mathrm{DR}_{1/2}(Y, X)),$$

as claimed. □

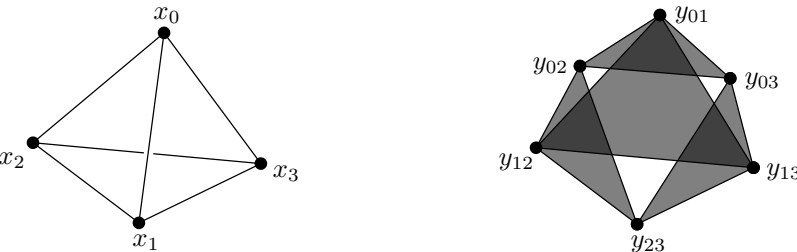

Figure 2: The complexes $\mathrm{D}_{1/2}(X, Y)$ (left) and $\mathrm{D}_{1/2}(Y, X)$ (right) from the proof of Proposition 4.4.

## 5 THE DOWKER-RIPS COMPLEX AS A DROP-IN REPLACEMENT FOR THE DOWKER COMPLEX

We now present a machine learning application in which using the Dowker-Rips complex instead of the Dowker complex leads to gains in speed while at the same time not negatively impacting performance. More concretely, it is shown in Stolz et al. (2024) that the Dowker complex may be used in a pipeline classifying tumor microenvironments into anti-tumor and pro-tumor macrophage dominant. We briefly review this pipeline here and refer the reader to Stolz et al. (2024, Section 5.1.1) for details.

First, (an image of) a tumor microenvironment is represented as a two-dimensional point cloud, each point of which is labeled according to whether it represents a blood vessel, necrotic cell, tumor cell or macrophage. Subsequently, the Dowker complex of one class of points relative to another is constructed; this is done for each of the label combinations macrophage-tumor, tumor-blood vessel and macrophage-blood vessel. For each of the complexes, persistent homology is computed, represented as a persistence diagram and discretized into a persistence image, yielding three persistence images, each of size $20 \times 20$ pixels, for each microenvironment. These persistence images are flattened into vectors, concatenated and passed to a support vector machine (SVM) for classification of the microenvironment into "anti-tumor" and "pro-tumor". As shown in Stolz et al. (2024, Section 5.1.3), this pipeline achieves a median classification accuracy of 86.6% across ten runs (controlling for randomized components in the SVM).

We reproduced the above pipeline and its result, and subsequently ran the same pipeline with the Dowker complex replaced by the Dowker-Rips complex; see Table 1 for the results.[2] In that table, we report the average classification accuracy with its standard deviation as well as the median accuracy across the ten runs.[3] We thus find that using the Dowker-Rips complex as a drop-in replacement for the Dowker complex in the pipeline above results in essentially the same classification performance. Crucially, however, we found that computation of the relevant complexes and their persistent homologies was sped up by a factor of over 14 when using the Dowker-Rips complex instead of the Dowker complex.[4]

Table 1: Results from microenvironment classification

| COMPLEX USED | MEAN ACCURACY | MEDIAN ACCURACY |
| --- | --- | --- |
| Dowker-Rips | 86.09±1.39 | 86.05 |
| Dowker | 85.69±1.49 | 85.51 |

For the above experiments, we implemented the Dowker-Rips complex as an open-source Python package compatible with the scikit-learn API. The reason for the speed gain of the Dowker-Rips complex over the Dowker complex stems from the fact that the former, unlike the latter, is a flag complex, and hence entirely determined by its 1-skeleton. This not only means that the Dowker-Rips complex is much less costly to construct than the Dowker complex, but also that its persistent homology can be computed using highly optimized state-of-the-art software. Indeed, in our implementation calculation of persistent homology is performed by ripser_parallel from the giotto-ph library (Pérez et al., 2021), which in turn is built on ripser (Bauer, 2021) and other software; both of these implementations are specifically adapted to flag complexes. In order to compute persistent homology of $DR_\bullet(X, Y)$ (where $X = \{x_1, \ldots, x_n\}$ and $Y = \{y_1, \ldots, y_m\}$ are subsets of $\mathbb{R}^N$ endowed with some distance function $d$), all that is needed is to create the matrix $M = \{m_{ij}\}_{i,j} \in \mathbb{R}^{n \times n}$ containing the filtration levels at which vertices and edges of $DR_\bullet(X, Y)$ appear. Letting $D = \{d(x_i, y_j)\}_{i,j} \in \mathbb{R}^{n \times m}$ denote the matrix of pairwise distances between $X$ and $Y$, the matrix $M$ may be obtained from $D$ by setting

- $m_{ii} := \min_k d(x_i, y_k)$, $1 \le i \le n$; and

- $m_{ij} := \min_k \max \{d(x_i, y_k), d(x_j, y_k)\}$, $1 \le i, j \le n$.

Passing $M$ to ripser_parallel then results in $PH_*(DR_\bullet(X, Y))$.

---

[2]Python code to run the pipelines is provided in the supplementary material for this submission. Running it requires our implementations of the Dowker-Rips and the Dowker complex, which are provided in the supplementary material as well.

[3]The discrepancy between the median accuracy of the pipeline using the Dowker complex reported in Table 1 and that found in Stolz et al. (2024) stems from the fact that we ported the original pipeline from Julia to Python.

[4]We ran our experiments on a laptop with a 12th Gen Intel Core i7-1260P processor running at 2.10GHz.

## REPRODUCIBILITY STATEMENT

All theoretical results are stated with complete proofs in the appendix. Definitions, assumptions, and intermediate lemmas are included to make the arguments self-contained. The supplementary material contains code that implements our method and experiments. The code is written in Python and depends only on standard libraries, or on libraries written by us that we provide in the supplementary material. Instructions for running the code and reproducing the results in the paper are included in the respective README files. Experiments can be reproduced on a standard laptop.

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

## A  APPENDIX

### A.1  PROOFS OF THEORETICAL RESULTS

In this section, we provide proofs for all theoretical results in the main text, separated according to which section the appear in the main text in. For convenience, we restate each result for before its proof.

#### A.1.1  PROOFS OF RESULTS PERTAINING TO MULTIPLICATIVE INTERLEAVINGS

**Theorem 1.6.** *Let $X, Y \subseteq Z$ where $(Z, d)$ is some metric space, and define the relations $R_\varepsilon \subseteq X \times Y$ by declaring $(x, y) \in R$ iff $d(x, y) \leq \varepsilon$ for $\varepsilon \geq 0$. Denote by $\mathrm{D}_\bullet(X, Y)$ the filtration given by $\{\mathrm{D}_{R_\varepsilon}(X, Y)\}_{\varepsilon \in \mathbb{R}^+}$, and similarly for $\mathrm{DR}_\bullet(X, Y)$. Then have that*

$$\mathrm{D}_\varepsilon(X, Y) \subseteq \mathrm{DR}_\varepsilon(X, Y) \subseteq \mathrm{D}_{3\varepsilon}(X, Y) \tag{2}$$

*for all $\varepsilon \geq 0$, and, in particular, that $\mathrm{D}_\bullet(X, Y)$ and $\mathrm{DR}_\bullet(X, Y)$ are multiplicatively 3-interleaved.*

*Proof.* It suffices to show that

$$D_\varepsilon(X, Y) \subseteq DR_\varepsilon(X, Y) \subseteq D_{3\varepsilon}(X, Y)$$

for all $\varepsilon \geq 0$; by defining $\varphi_\varepsilon$ and $\psi_\varepsilon$ as inclusions, the commutativity of the required diagrams then follows immediately.

Let $\varepsilon \geq 0$. The inclusion $D_\varepsilon(X, Y) \subseteq DR_\varepsilon(X, Y)$ is immediate from the definition of $DR_\varepsilon(X, Y)$ as the flagification of $D_\varepsilon(X, Y)$.

Suppose now that $DR_\varepsilon(X, Y)$ contains some simplex $\sigma = [x_0, \dots, x_n]$, where $x_0, \dots, x_n \in X$. By definition, this means that for any $x_i, x_j \in \sigma$ there exists an element $y_{ij} \in Y$ such that $d(x_i, y_{ij}) \leq \varepsilon$ and $d(x_j, y_{ij}) \leq \varepsilon$. Now, given any $x_i \in \sigma$, we have that

$$\begin{aligned}
d(x_i, y_{kl}) &\leq d(x_i, x_k) + d(x_k, y_{kl}) \\
&\leq d(x_i, y_{ki}) + d(y_{ki}, x_k) + d(x_k, y_{kl}) \\
&\leq 3\varepsilon
\end{aligned}$$

for any $0 \leq k < j \leq n$. Hence $\sigma \in D_{3\varepsilon}(X, Y)$, as claimed. $\square$

**Theorem 1.7.** *Let $X, Y \subseteq Z$ where $(Z, d)$ is some metric space, and define the relations $R_\varepsilon \subseteq X \times Y$ as in Theorem 1.6, $\varepsilon \geq 0$. Denote by $DR_\bullet(X, Y)$ the filtration given by $\{DR_{R_\varepsilon}(X, Y)\}_{\varepsilon \in \mathbb{R}^+}$, and similarly for $DR_\bullet(Y, X)$. Then $DR_\bullet(X, Y)$ and $DR_\bullet(Y, X)$ are multiplicatively $3$-interleaved.*

*Proof.* Consider the following chain of maps

$$DR_\varepsilon(X, Y) \xrightarrow{\iota_{DR,D}^\varepsilon} D_{3\varepsilon}(X, Y) \xrightarrow{\iota^{(1)}} D_{3\varepsilon}^{(1)}(X, Y) \xrightarrow{\Gamma} D_{3\varepsilon}(Y, X) \xrightarrow{\iota_{D,DR}^{3\varepsilon}} DR_{3\varepsilon}(Y, X),$$

where $\iota_{DR,D}^\varepsilon$ and $\iota_{D,DR}^{3\varepsilon}$ denote the inclusion maps from Theorem 1.6, $\iota^{(1)}$ denotes the inclusion of the respective complex into its first barycentric subdivision, and where $\Gamma$ denotes the simplicial map from Chowdhury & Mémoli (2018). We define $\varphi_\varepsilon := \iota_{D,DR}^{3\varepsilon} \circ \Gamma \circ \iota^{(1)} \circ \iota_{DR,D}^\varepsilon$. The functions $\psi_\varepsilon$ are defined symmetrically.

Consider first the following diagram:

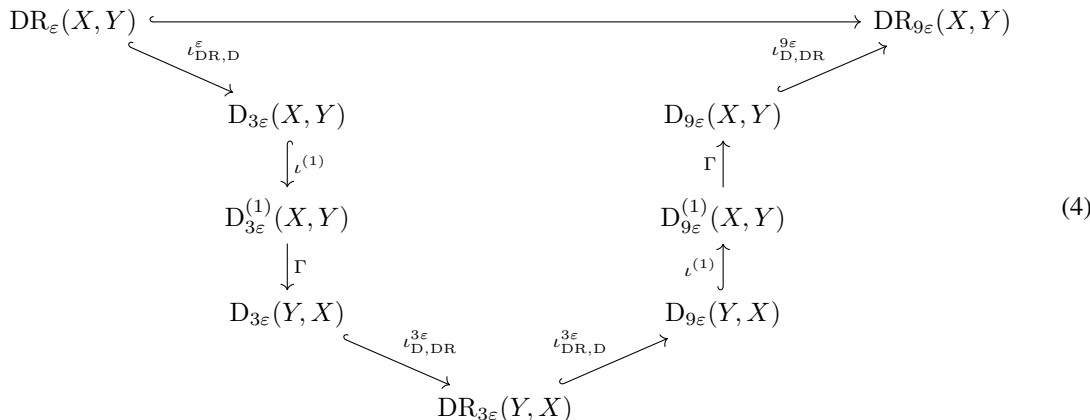

(4)

By definition of $\varphi_\varepsilon$ and $\psi_\varepsilon$, this is exactly the triangular diagram required for multiplicative interleavings. It follows from functoriality of $\Gamma$ established in Chowdhury & Mémoli (2018) together with the fact that all other maps are inclusion maps that this diagram commutes.

Similarly, the relevant trapezoidal diagram is the following:

$$
\begin{array}{ccc}
\mathrm{DR}_\varepsilon(X,Y) & \longrightarrow & \mathrm{DR}_{\varepsilon+\varepsilon'}(X,Y) \\
\ \downarrow{\scriptstyle \iota^{\varepsilon}_{\mathrm{DR,D}}} & & \ \downarrow{\scriptstyle \iota^{\varepsilon+\varepsilon'}_{\mathrm{DR,D}}} \\
\mathrm{D}_{3\varepsilon}(X,Y) & & \mathrm{D}_{3(\varepsilon+\varepsilon')}(X,Y) \\
\ \downarrow{\scriptstyle \iota^{(1)}} & & \ \downarrow{\scriptstyle \iota^{(1)}} \\
\mathrm{D}^{(1)}_{3\varepsilon}(X,Y) & & \mathrm{D}^{(1)}_{3(\varepsilon+\varepsilon')}(X,Y) \qquad (5) \\
\ \downarrow{\scriptstyle \Gamma} & & \ \downarrow{\scriptstyle \Gamma} \\
\mathrm{D}_{3\varepsilon}(Y,X) & & \mathrm{D}_{3(\varepsilon+\varepsilon')}(Y,X) \\
\ \downarrow{\scriptstyle \iota^{3\varepsilon}_{\mathrm{D,DR}}} & & \ \downarrow{\scriptstyle \iota^{3(\varepsilon+\varepsilon')}_{\mathrm{D,DR}}} \\
\mathrm{DR}_{3\varepsilon}(X,Y) & \longrightarrow & \mathrm{DR}_{3(\varepsilon+\varepsilon')}(X,Y)
\end{array}
$$

Again, this diagram commutes by functoriality of $\Gamma$ and the fact that all other maps are inclusion maps. $\square$

### A.1.2 PROOFS OF RESULTS PERTAINING TO DOWKER-RIPS DUALITY

**Lemma 4.2.** *The homotopy equivalence* $\psi\colon |\mathrm{D}^{(1)}_R(X,Y)| \to |\mathrm{D}_R(Y,X)|$ *extends to a continuous map*

$$
\varphi\colon |\mathcal{F}^k(\mathrm{D}_R(X,Y))^{(1)}| \to |\mathcal{F}^k(\mathrm{D}_R(Y,X))|
$$

*for any* $k \geq 2$.

*Proof.* To prove the lemma, we must define $\varphi$ on the portion of $|\mathcal{F}^k(\mathrm{D}_R(X,Y))^{(1)}|$ that is not present in $|\mathrm{D}^{(1)}_R(X,Y)|$. This portion consists of the geometric realizations of those simplices that belong to $\mathcal{F}^k(\mathrm{D}_R(X,Y))$, but not to $\mathrm{D}_R(X,Y)$. Let $\sigma \in \mathcal{F}^k(\mathrm{D}_R(X,Y)) \setminus \mathrm{D}_R(X,Y)$ be such a simplex. Since $\sigma$ is $k$-dimensional, we may write $\sigma = [x_0,\dots,x_k]$ for some $x_0,\dots,x_k \in X$. Moreover, by definition of $\mathcal{F}^k(\mathrm{D}_R(X,Y))$, it must be the case that all proper faces of $\sigma$ belong to $\mathrm{D}_R(X,Y)$. Letting $\mathcal{I}_k$ denote the set of subsets $I \subseteq \{0,\dots,k\}$ such that $0 < |I| < k+1$, we thus have that $[x_i]_{i \in I} \in \mathrm{D}_R(X,Y)$ for all $I \in \mathcal{I}_k$. Given $I \in \mathcal{I}_k$, let $x_I \in \mathrm{D}^{(1)}_R(X,Y)$ denote the vertex corresponding to the face $[x_i]_{i \in I}$ of $\sigma$, and define the subcomplex $C^X_{\partial\sigma} \subseteq \mathrm{D}^{(1)}_R(X,Y)$ as the barycentric subdivision of the complex consisting of the proper faces of $\sigma$. Similarly, define $C^X_\sigma \subseteq \mathcal{F}^k(\mathrm{D}_R(X,Y))^{(1)}$ as the barycentric subdivision of $\sigma$. See Figure 3a for a schematic illustration of $C^X_{\partial\sigma}$ and $C^X_\sigma$ in the case where $k = 2$.

Given any $I \in \mathcal{I}_k$, set $y_I := \Gamma(x_I) \in \mathrm{D}_R(Y,X)$. Note that a collection of these elements spans a simplex $[y_{I_1},\dots,y_{I_l}] \in \mathrm{D}_R(Y,X)$ whenever $I_1,\dots,I_l \in \mathcal{I}_k$ are such that $I_1 \cap \cdots \cap I_l \neq \varnothing$. To see this, let $I_1,\dots,I_l \in \mathcal{I}_k$ be such sets. Then, by definition of $\Gamma$, we have that $(x_i, y_{I_1}),\dots,(x_i, y_{I_l}) \in R$ for all $i \in I_1 \cap \cdots \cap I_l$, and hence that $[y_{I_1},\dots,y_{I_l}] \in \mathrm{D}_R(Y,X)$.[5] In particular, we have that $\mathrm{D}_R(Y,X)$ contains the $k+1$ simplices $[y_I]_{\{I \in \mathcal{I}_k | i \in I, |I| = k\}}$, each of dimension $k-1$, for all $i = 0,\dots,k$. Hence $\mathcal{F}^k(\mathrm{D}_R(Y,X))$ contains the $k$-dimensional simplex $[y_I]_{\{I \in \mathcal{I}_k | |I| = k\}}$. With this at hand, define the subcomplex $C^Y_{\partial\sigma} \subseteq \mathrm{D}_R(Y,X)$ as having vertex set $\{y_I \mid I \in \mathcal{I}_k\}$ and simplices $[y_{I_1},\dots,y_{I_l}]$, for $I_1,\dots,I_l \in \mathcal{I}_k$ such that $I_1 \cap \cdots \cap I_l \neq \varnothing$. Furthermore, define $C^Y_\sigma \subseteq \mathcal{F}^k(\mathrm{D}_R(Y,X))$ to be the complex obtained from $C^Y_{\partial\sigma}$ by adding the simplex $[y_I]_{\{I \in \mathcal{I}_k | |I| = k\}}$. See Figure 3b for a schematic illustration of $C^Y_{\partial\sigma}$ and $C^Y_\sigma$ in the case where $k = 2$.

By construction, we have that $\Gamma(C^X_{\partial\sigma}) \subseteq C^Y_{\partial\sigma}$, and hence, by passing to geometric realizations, that $\psi(|C^X_{\partial\sigma}|) \subseteq |C^Y_{\partial\sigma}| \subseteq |C^Y_\sigma|$. It remains to show that $\psi$ extends from $|C^X_{\partial\sigma}|$ to $|C^X_\sigma|$, for which, in turn,

---

[5] Note that the elements $y_I \in Y$ for $I \in \mathcal{I}_k$ are not necessarily pairwise distinct: if $I, J \in \mathcal{I}_k$ are such that $I \subseteq J$, it can be the case that $y_I = y_J \in Y$, in which case the edge $[y_I, y_J]$ degenerates to a point.

it suffices to show that $|C_\sigma^Y|$ is contractible (see, e.g., Hatcher (2002, Corollary 4.73)). To that end, observe that for any $i = 0, \ldots, k$, the simplex $[y_I]_{\{I \in \mathcal{I}_k | i \in I\}} \in C_\sigma^Y$, that is, the simplex induced by all $y_I$ whose subscript contains $i$, is a maximal face of $C_\sigma^Y$. Indeed, $[y_I]_{\{I \in \mathcal{I}_k | i \in I\}}$ is the only maximal face containing the vertex $y_i$, and hence the latter vertex is a free face of $C_\sigma^Y$. We may thus collapse $C_\sigma^Y$ with respect to the free faces $y_0, \ldots, y_k$, which results in a complex homotopy equivalent to $C_\sigma^Y$. This resulting complex is the subcomplex of $C_\sigma^Y$ induced by the vertices $y_I$ for $I \in \mathcal{I}_k$ and $|I| > 1$. Similarly to before, all vertices of this new complex that are of the form $y_I$ for $I \in \mathcal{I}_k$ and $|I| = 2$ are free faces. We may thus collapse this complex with respect to these free faces to obtain a complex that is still homotopy equivalent to $C_\sigma^Y$. Repeating this process eventually results in the subcomplex of $C_\sigma^Y$ induced by the vertices $y_I$ for $I \in \mathcal{I}_k$ and $|I| = k$, and demonstrates that this resulting complex is homotopy equivalent to the original complex $C_\sigma^Y$. As we have seen in the previous paragraph, we have that $[y_I]_{\{I \in \mathcal{I}_k | |I| = k\}} \in \mathcal{F}^k(\mathrm{D}_R(Y, X))$. In other words, the complex resulting from iteratively collapsing as above is simply a $k$-dimensional simplex and hence $C_\sigma^Y$, being homotopy equivalent to a simplex, is contractible. $\qquad\square$

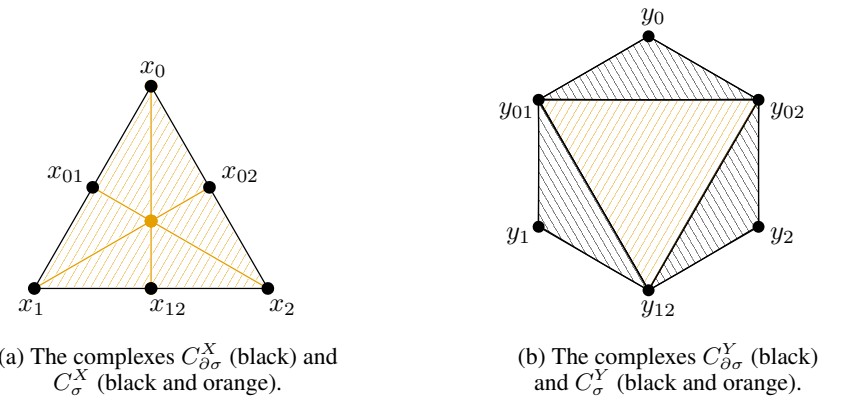

(a) The complexes $C_{\partial\sigma}^X$ (black) and $C_\sigma^X$ (black and orange).

(b) The complexes $C_{\partial\sigma}^Y$ (black) and $C_\sigma^Y$ (black and orange).

Figure 3: Schematics accompanying the proof of Lemma 4.2 for the case where $k = 2$.

**Proposition 4.3.** *Let $X$ and $Y$ be two finite sets, let $R \subseteq R' \subseteq X \times Y$ be two non-empty relations, and let $k \geq 2$ an integer. Then there exist continuous maps $\varphi \colon |\mathcal{F}^k(\mathrm{D}_R(X, Y))| \to |\mathcal{F}^k(\mathrm{D}_R(Y, X))|$ and $\varphi' \colon |\mathcal{F}^k(\mathrm{D}_{R'}(X, Y))| \to |\mathcal{F}^k(\mathrm{D}_{R'}(Y, X))|$ that induce isomorphisms on the level of $i$-dimensional homology for $i = 0, \ldots, k - 1$, and, moreover, such that the diagram*

$$
\begin{array}{ccc}
|\mathcal{F}^k(\mathrm{D}_R(X, Y))| & \lhook\joinrel\longrightarrow & |\mathcal{F}^k(\mathrm{D}_{R'}(X, Y))| \\
\varphi \downarrow & & \downarrow \varphi' \\
|\mathcal{F}^k(\mathrm{D}_R(Y, X))| & \lhook\joinrel\longrightarrow & |\mathcal{F}^k(\mathrm{D}_{R'}(Y, X))|
\end{array}
\tag{3}
$$

*commutes up to homotopy. Here, the horizontal maps are given by inclusion.*

*Proof.* Let $\varphi \colon |\mathcal{F}^k(\mathrm{D}_R(X, Y))^{(1)}| \to |\mathcal{F}^k(\mathrm{D}_R(Y, X))|$ be an extension of the homotopy equivalence $\psi \colon |\mathrm{D}_R^{(1)}(X, Y)| \to |\mathrm{D}_R(Y, X)|$, whose existence is guaranteed by Lemma 4.2.

We first show that $\varphi$ induces isomorphisms on the level of $i$-dimensional homology for $i = 0, \ldots, k - 1$. To that end, consider the commutative diagram

$$
\begin{array}{ccc}
|\mathrm{D}_R(X, Y)| & \overset{\iota^X}{\lhook\joinrel\longrightarrow} & |\mathcal{F}^k(\mathrm{D}_R(X, Y))| \\
\psi \downarrow & & \downarrow \varphi \\
|\mathrm{D}_R(Y, X)| & \overset{\iota^Y}{\lhook\joinrel\longrightarrow} & |\mathcal{F}^k(\mathrm{D}_R(Y, X))|
\end{array}
$$

where $\iota^X$ and $\iota^Y$ denote inclusion maps, and where we identified $|\mathrm{D}_R^{(1)}(X,Y)|$ and $|\mathrm{D}_R(X,Y)|$ via the canonical homeomorphism between them. Now, since $|\mathcal{F}^k(\mathrm{D}_R(Y,X))|$ is obtained from $|\mathrm{D}_R(Y,X)|$ by attaching $k$-dimensional cells, it follows that $\iota^Y$ induces an isomorphism on the level of $i$-dimensional homology for $i = 0, \ldots, k-2$, and a surjection on the level of $(k-1)$-dimensional homology. Hence, using the fact that $\psi$ is a homotopy equivalence, we have that the map $\iota^Y \circ \psi$ induces a surjection on the level of $i$-dimensional homology for $i = 0, \ldots, k-1$. By commutativity of the above diagram, the same is true about the map $\varphi \circ \iota^X$, and hence the map that $\varphi$ alone induces on the level of $i$-dimensional homology must be a surjection, too, for $i = 0, \ldots, k-1$. Swapping the roles of $X$ and $Y$ in the above, it follows that $\mathrm{H}_i(|\mathcal{F}^k(\mathrm{D}_R(X,Y))|)$ surjects onto $\mathrm{H}_i(|\mathcal{F}^k(\mathrm{D}_R(Y,X))|)$ and vice versa for $i = 0, \ldots, k-1$. Since all simplicial complexes involved are finite, we thus have that $\mathrm{H}_i(|\mathcal{F}^k(\mathrm{D}_R(X,Y))|) \cong \mathrm{H}_i(|\mathcal{F}^k(\mathrm{D}_R(Y,X))|)$, and hence that $\varphi$ induces isomorphisms on the level of $i$-dimensional homology for $i = 0, \ldots, k-1$, as claimed.

To prove commutativity of Diagram 3 in the statement of Proposition 4.3, consider the following diagram

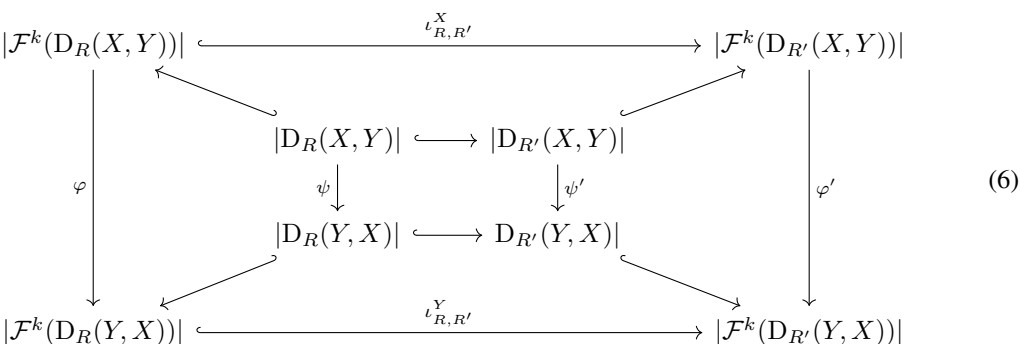

$$\text{(6)}$$

where $\varphi$ and $\varphi'$ are extensions of the homotopy equivalences $\psi$ and $\psi'$, respectively, as before; where hooked arrows denote inclusion maps; and where we identified $|\mathrm{D}_R(X,Y)|$ and $|\mathrm{D}_R(X,Y)^{(1)}|$ as before. Observe that the upper and lower trapezoids are commutative because the respective maps are inclusion maps, while commutativity of the left and right trapezoids follows from the fact that $\varphi$ and $\varphi'$ are extensions of $\psi$ and $\psi'$, respectively. Moreover, the inner rectangle commutes up to homotopy by Chowdhury & Mémoli (2018, Theorem 3) and we may thus assume its precise commutativity.[6]

Now, let $x \in |\mathcal{F}^k(\mathrm{D}_R(X,Y))|$. If $x \in |\mathrm{D}_R(X,Y)| \subseteq |\mathcal{F}^k(\mathrm{D}_R(X,Y))|$, then the fact that $(\iota_{R,R'}^Y \circ \varphi)(x) = (\varphi' \circ \iota_{R,R'}^X)(x)$ is an immediate consequence of commutativity of the trapezoids and the inner rectangle in Diagram 6. Suppose now that $x \in |\mathcal{F}^k(\mathrm{D}_R(X,Y))| \setminus |\mathrm{D}_R(X,Y)|$, so that $x$ belongs to the geometric realization of some simplex $\sigma_x$ that is present in $\mathcal{F}^k(\mathrm{D}_R(X,Y))$ but not in $\mathrm{D}_R(X,Y)$. Note that the extensions $\varphi$ and $\varphi'$ are constructed from $\psi$ and $\psi'$, respectively, on a per simplex basis. We may thus assume that $\varphi'$ agrees with $\varphi$ on the geometric realizations of simplices stemming that are already present in $\mathcal{F}^k(\mathrm{D}_R(X,Y))$, which establishes the equality $(\iota_{R,R'}^Y \circ \varphi)(x) = (\varphi' \circ \iota_{R,R'}^X)(x)$ in this case. $\qquad\square$

**Theorem 1.8.** *Let $X$ and $Y$ be two finite sets and let $\{R_j\}_{j \in J}$ be a sequence of relations such that $R_j \subseteq X \times Y$ for all $j \in J$, and $R_j \subseteq R_{j'}$ whenever $j \leq j'$, where $J$ is some totally ordered index set. Given an integer $k \geq 2$, denote by $\mathcal{F}^{\geq k}(\mathrm{D}_\bullet(X,Y))$ the filtration given by $\big\{\mathcal{F}^{\geq k}(\mathrm{D}_{R_j}(X,Y))\big\}_{j \in J}$, and similarly for $\mathcal{F}^{\geq k}(\mathrm{D}_\bullet(Y,X))$. Then we have that*

$$\mathrm{PH}_i(\mathcal{F}^{\geq k}(\mathrm{D}_\bullet(X,Y))) \cong \mathrm{PH}_i(\mathcal{F}^{\geq k}(\mathrm{D}_\bullet(Y,X)))$$

*for $i = 0, \ldots, k-1$.*

---

[6]Precise commutativity of this rectangle is achieved by making the choices of $y_\sigma$ in the definition of the maps $\psi$ and $\psi'$ in a consistent manner.

*Proof of Theorem 1.8.* Let $j, j' \in J$ be such that $j < j'$, and consider the following diagram of maps

$$
\begin{array}{ccc}
|\mathcal{F}^{\geq k}(\mathrm{D}_{R_j}(X,Y))| & \longhookrightarrow & |\mathcal{F}^{\geq k}(\mathrm{D}_{R_{j'}}(X,Y))| \\
\uparrow & & \uparrow \\
|\mathcal{F}^{k}(\mathrm{D}_{R_j}(X,Y))| & \longhookrightarrow & |\mathcal{F}^{k}(\mathrm{D}_{R_{j'}}(X,Y))| \\
\varphi\downarrow & & \downarrow\varphi' \\
|\mathcal{F}^{k}(\mathrm{D}_{R_j}(Y,X))| & \longhookrightarrow & |\mathcal{F}^{k}(\mathrm{D}_{R_{j'}}(Y,X))| \\
\downarrow & & \downarrow \\
|\mathcal{F}^{\geq k}(\mathrm{D}_{R_j}(Y,X))| & \longhookrightarrow & |\mathcal{F}^{\geq k}(\mathrm{D}_{R_{j'}}(Y,X))|
\end{array}
\tag{7}
$$

where $\varphi$ and $\varphi'$ are maps as in the statement of Proposition 4.3 and where hooked arrows denote inclusion maps. The top and bottom rectangles are commutative since the maps involved are inclusion maps, and commutativity of the middle rectangle follows Proposition 4.3.

Since, for instance, $\mathcal{F}^{\geq k}(\mathrm{D}_{R_j}(X,Y))$ and $\mathcal{F}^{k}(\mathrm{D}_{R_j}(X,Y))$ share the same $k$-skeleton, it follows that the top left inclusion map induces an isomorphism on the level of $i$-dimensional homology for $i = 0, \ldots, k-1$. Similarly, it follows that the same is true for the other vertical inclusion maps, and hence, by Proposition 4.3, for all vertical maps. Applying the homology functor to Diagram 7, and suppressing the two middle rows, we obtain the commutative diagram

$$
\begin{array}{ccc}
\mathrm{H}_i(|\mathcal{F}^{\geq k}(\mathrm{D}_{R_j}(X,Y))|) & \longrightarrow & \mathrm{H}_i(|\mathcal{F}^{\geq k}(\mathrm{D}_{R_{j'}}(X,Y))|) \\
\cong\updownarrow & & \cong\updownarrow \\
\mathrm{H}_i(|\mathcal{F}^{\geq k}(\mathrm{D}_{R_j}(Y,X))|) & \longrightarrow & \mathrm{H}_i(|\mathcal{F}^{\geq k}(\mathrm{D}_{R_{j'}}(Y,X))|)
\end{array}
\tag{8}
$$

for $i = 0, \ldots, k-1$. Diagram 8 thus establishes an isomorphism of persistence modules $\mathrm{PH}_i(\mathcal{F}^{\geq k}(\mathrm{D}_\bullet(X,Y))) \cong \mathrm{PH}_i(\mathcal{F}^{\geq k}(\mathrm{D}_\bullet(Y,X)))$ for $i = 0, \ldots, k-1$, as claimed. $\qquad\square$

**Theorem 1.10.** *Let $(Z, d)$ be a metric space and let $X, Y \subseteq Z$ be non-empty and finite disjoint subsets. For $\varepsilon \geq 0$, define the relation $R_\varepsilon \subseteq X \times Y$ by*

$$(x, y) \in R_\varepsilon \quad \text{iff} \quad d(x, y) \leq \varepsilon.$$

*Denote by $\mathrm{DR}_\bullet(X, Y)$ the filtration given by $\{\mathrm{DR}_{R_\varepsilon}(X, Y)\}_{\varepsilon \in \mathbb{R}^+}$, and similarly for $\mathrm{DR}_\bullet(Y, X)$. Then we have that*

$$\mathrm{PH}_i(\mathrm{DR}_\bullet(X, Y)) \cong \mathrm{PH}_i(\mathrm{DR}_\bullet(Y, X))$$

*for $i = 0, 1$.*

*Proof.* This is an immediate consequence of setting $k = 2$ in Theorem 1.8. $\qquad\square$

