# OpenReview forum: "Flagifying the Dowker Complex"
_ICLR.cc/2026/Conference — ICLR 2026 Conference Withdrawn Submission_

### Official Review · Reviewer_4Wev · 2025-10-27

**Soundness:** 4
**Presentation:** 3
**Contribution:** 3
**Rating:** 6
**Confidence:** 5

**Summary:**

This paper introduces the Dowker-Rips complex (DRR), a flagified version of the Dowker complex, as a computationally cheaper alternative for applications in topological data analysis (TDA). The authors provide strong theoretical foundations for DRR, including a tight multiplicative 3-interleaving relationship with the original Dowker complex and a partial preservation of Dowker duality (isomorphism of homology groups in dimensions 0 and 1). An application to tumour microenvironment classification demonstrates that DRR can replace the Dowker complex in practice, achieving over 14x speedup with no significant loss in classification performance. However, the paper’s empirical evaluation is limited to a single use case, and the failure of duality in higher homological dimensions is only explored through a specific counterexample without deeper structural analysis. Additionally, the transition from theoretical results to practical application could be more clearly integrated for broader ML audiences.

**Strengths:**

Novel theoretical definition of Dowker-Rips complex as a flagified version of Dowker complex.

Tight 3-interleaving bound formally quantified and proven.

Partial Dowker duality result, which is both technically interesting and practically useful for computational savings.

Clean application to a biomedical ML problem showing real computational benefits (14x speedup) without hurting accuracy.

**Weaknesses:**

Empirical evaluation is limited to a single application domain (tumor microenvironments). Additional experiments on synthetic datasets or diverse data types (e.g., higher-dimensional manifolds, graphs, or asymmetric relations in NLP) would better support the generalizability of the proposed method.

The failure of Dowker duality in higher homological dimensions is only illustrated via a specific counterexample (Proposition 4.4). A deeper exploration, e.g., necessary or sufficient conditions for duality to fail, is missing.

Accessibility is somewhat limited for readers outside of TDA. While the math is correct, the exposition assumes substantial background in algebraic topology, which could limit adoption or understanding among ML researchers.

The paper does not sufficiently position DRR relative to related TDA methods that also aim for computational efficiency, such as Sparse Dowker Nerves or alternative nerve constructions, nor does it clearly explain distinctions in scope or advantage.

**Questions:**

Could the authors clarify whether DRR maintains stability guarantees (e.g., bottleneck distance bounds) when compared to Dowker complexes under metric perturbations?

Is there a potential to extend partial duality to H₂ under additional structural assumptions on the data or relation
𝑅?

The application focuses on 2D data. Have the authors tried DRR on higher-dimensional embeddings or manifolds (e.g., word embeddings, graphs)?

Can the interleaving constant of 3 be reduced in special cases (e.g., when X and Y are highly regular or symmetric)?

Are there known failure modes where DRR significantly distorts persistence (e.g., false positives in H₁ loops)?

---

> ### Author Response · Authors · 2025-11-19
> **Retraction**
>
> Given that all reviewers have expressed concern about ICLR being the right venue for our paper, we have decided to retract our paper.
> Nevertheless, we would like to thank the reviewer for taking the time to provide constructive feedback.

---

### Official Review · Reviewer_HgSP · 2025-10-30

**Soundness:** 4
**Presentation:** 3
**Contribution:** 3
**Rating:** 2
**Confidence:** 4

**Summary:**

This paper proposes a novel Topological Data Analysis (TDA) filtration for a
pair point cloud with a (persistent) relation, based on the Dowker
complex/filtration.

After introducing the contributions, the authors detail the necessary
background, and present the novel filtration, that can be built from
flagify-ing the already known Dowker filtration on its 1-skeleton, i.e., all
$d\ge 2$-simplices' filtration values are changed to their minimum possible
value (w.r.t. the 1-skeleton).
This results in a filtration that still preserve satisfying guarantees, e.g.,
bounded multiplicative interleaving error, with sharp bounds, and (persistence)
relation duality guarantees.

The paper concludes with an experimental section, showcasing the statistical
performance of the DowkerRips complex when compared against the usual Dowker
complex.

**Strengths:**

- Math seems to be sound, intuitive, with sharp bounds / examples.
  - Efficient algorithms are nice.
  - Code is available, documented and scikit-compatible. Furthermore, it relies
  on efficient subroutines of the Gudhi/numba libraries, which should result in
  an efficient implementation.

**Weaknesses:**

- Scope. I'm not sure that this conference is the right scope for this paper.
 I'm not sure this work is easy to read for the machine learning community.
 Furthermore, the main strenghts of this paper are not in machine learning /
 representation learning.
- Experimental section.
  - Dataset.
  - Only one dataset is used.
  - DR having better performance than D is confusing.
  - The dataset is small: ~1000 point clouds of size 200-900.
  - No real competitors. I agree that it may not be "fair" to compare to SOTA
  deep learning approach, but a comparison with at least other TDA approaches
  would be interesting.
  - The hyperparameters can only be found in the code (e.g., in the
  `compute_svm_accuracies.py` file).
  - If I understand correctly the motivation for this filtration, it is meant
  to efficiently approximate a Dowker filtration.
  - How does Dowker vs Dowker Rips scale w.r.t. the number of points?
      - What are the theoretical computational complexities?

Linewise comments:
- l176-182. DR and D share the same 1-skeleton, so this argument can only apply
to the homology of degree 1. This is also at the price of a 3 multiplicative
error (or a bit less on Euclidean data).

**Questions:**

see weaknesses.
 - On Euclidean data, can this construction be related / adapted to chromatic alpha complexes ?

---

> ### Author Response · Authors · 2025-11-19
> **Retraction**
>
> Given that all reviewers have expressed concern about ICLR being the right venue for our paper, we have decided to retract our paper.
> Nevertheless, we would like to thank the reviewer for taking the time to provide constructive feedback.

---

### Official Review · Reviewer_G75M · 2025-10-31

**Soundness:** 4
**Presentation:** 4
**Contribution:** 3
**Rating:** 6
**Confidence:** 4

**Summary:**

Dowker complex is a well-known construction that produces a simplicial complex given a binary relation. The paper proposes to "relax" it in the similar way Vietoris--Rips is relaxing Cech, namely, by "flagification": take a Dowker complex and for a fixed integer $k$,  add a simplex to the $k$-flagification, if all its $k$-dimensional faces are in the original Dowker complex. For $k=2$, this means that we add a simplex, if all its edges are present.
To get some approximation guarantees in the sense of persistent homology, one needs a filtration, so one needs a family of relations. The results of the paper (multiplicative interleaving and Dowker duality in dimensions $0..k$) are for the relations $R_\epsilon = \set{ (x, y) \in X \times Y : d(x, y) \leq \epsilon}$. The paper proves the tight bounds on how far apart the Dowker filtration and the Dowker--Rips filtration are topologically and proves that the Dowker duality holds in dimensions 0 and 1 (as fairly pointed out, the most often used in practice).
The paper concludes with one experiment for tumor classification.

From the TDA viewpoint, the contributions of the paper are valuable, and if I rated the same submission for a computational topology venue, I would say it's 'accept', not 'weak accept'.

**Strengths:**

- The theoretical results of the paper are a valuable contribution. Taking the flag complex of Dowker--Rips is a very natural choice from the TDA perspective, and the analysis in the paper is complete: there are approximation guarantees for this approach plus counterexamples demonstrating their tightness.
- The paper is well-organized and well-written.
- The authors also provide code for practical usage of this approach.

**Weaknesses:**

- The reviewer is unsure if this paper is a good fit for an ML conference. The main bulk of the paper is mathematical, and the TDA audience would certainly be interested in it, but there is only one ML example in the end.

**Questions:**

Dowker complexes are defined for arbitrary relations. While the "being $\varepsilon$-close to a point in $Y$" is a relation that makes a lot of sense and, probably, has the most potential applications, are there other interesting relations that can give rise to a Dowker-like filtration that can be flagified?

---

> ### Author Response · Authors · 2025-11-19
> **Retraction**
>
> Given that all reviewers have expressed concern about ICLR being the right venue for our paper, we have decided to retract our paper.
> Nevertheless, we would like to thank the reviewer for taking the time to provide constructive feedback.

---

### Note · Authors · 2025-11-19

I have read and agree with the venue's withdrawal policy on behalf of myself and my co-authors.